# Recognition and Scoring Physical Exercises via Temporal and Relative Analysis of Skeleton Nodes Extracted from the Kinect Sensor

**DOI:** 10.3390/s24206713

**Published:** 2024-10-18

**Authors:** Raana Esmaeeli, Mohammad Javad Valadan Zoej, Alireza Safdarinezhad, Ebrahim Ghaderpour

**Affiliations:** 1Department of Photogrammetry and Remote Sensing, Faculty of Surveying Engineering, K. N. Toosi University of Technology, Tehran 19967-15443, Iran; r.esmaeeli@email.kntu.ac.ir (R.E.); valadanzouj@kntu.ac.ir (M.J.V.Z.); 2Department of Geodesy and Surveying Engineering, Tafresh University, Tafresh 39518-79611, Iran; 3Department of Earth Sciences, Sapienza University of Rome, Ple Aldo Moro, 5, 00185 Rome, Italy

**Keywords:** human activity recognition, depth sensor, Kinect, mutual information, constrained energy minimization, scoring, physical exercise

## Abstract

Human activity recognition is known as the backbone of the development of interactive systems, such as computer games. This process is usually performed by either vision-based or depth sensors. So far, various solutions have been developed for this purpose; however, all the challenges of this process have not been completely resolved. In this paper, a solution based on pattern recognition has been developed for labeling and scoring physical exercises performed in front of the Kinect sensor. Extracting the features from human skeletal joints and then generating relative descriptors among them is the first step of our method. This has led to quantification of the meaningful relationships between different parts of the skeletal joints during exercise performance. In this method, the discriminating descriptors of each exercise motion are used to identify the adaptive kernels of the Constrained Energy Minimization method as a target detector operator. The results indicated an accuracy of 95.9% in the labeling process of physical exercise motions. Scoring the exercise motions was the second step after the labeling process, in which a geometric method was used to interpolate numerical quantities extracted from descriptor vectors to transform into semantic scores. The results demonstrated the scoring process coincided with the scores derived by the sports coach by a 99.5 grade in the R^2^ index.

## 1. Introduction

Computer games and entertainment have evolved in today’s world. The control of games is done through physical interaction instead of joysticks [1,2]. Virtual reality glasses and 3D sensors, such as Kinect, are examples of hardware used for controlling interactive games [3,4]. Alongside advancements in the gaming and entertainment industry, 3D sensors and virtual reality glasses are also being utilized as tools for smart classrooms, the metaverse, virtual twins, medical science, automated guided vehicles, and 3D modeling [5,6,7,8,9,10,11,12,13,14,15,16].

Kinect is a sensor that captures red–green–blue (RGB) and depth videos simultaneously. The depth data are generated using Time of Flight (TOF) detectors or by matching projected NIR patterns and photogrammetric triangulations, depending on the version [17,18]. Accordingly, the digital depth data numbers are either the quantized values of TOF measurements or disparities. These values have an inverse relation with distances between objects and the sensor [17]. The depth videos have been a valuable data source for various applications. Examples are fusing depth images to complete 3D models [19], positioning virtual objects within captured scenes [20], interior layout design [21], virtual clothes put on [22], the creation of interactive classrooms [4], and motion recognition [1].

Real-time control of computer games or automatic evaluation of dance or physical exercises via depth videos are applications that require the initial identification and extraction of the human body and its skeletal structure [4,23,24]. Skeletal structure extraction involves identifying at least 20 3D joint points of the body, as shown in Figure 1, from each depth video frame. Tracking these points can help distinguish human actions and movements.

Several methods have been developed to recognize human body parts and extract their skeletal structures from either RGB or depth images. While the methods using single RGB images can usually achieve 2D body parts estimation without or with weak depth information, which limits the performance of human action analysis, depth image-based methods could overcome more of these deficiencies. For example, Sideridou et al. [25] used RGB videos captured by smartphones and the BlazePose body pose estimation model for assessing exercises. Due to the uncertainties of the depth component of the extracted body parts, they were forced to change the viewpoint of the video camera to record the exercises that were not reliable when captured from other viewpoints [25]. Hence, human 3D skeleton detection from depth videos will probably be a better choice for human action recognition purposes. Shoton et al. [26] detected human body parts in depth images using a trained Random Forest segmentation, dividing them into 31 contiguous pieces, some of which were merged before identifying the skeletal points. In this method, the body’s joint locations are identified by calculating the centroid of each segment using the moving average method with the Gaussian kernel. In general, the accuracy and quality of methods for identifying skeletal points depend on the level of noise in the depth images and the level of body occlusions. Yeung et al. [27] attempted to solve the challenges of occlusion in extracting 3D body skeletal structures from single-view depth images by simultaneously using two Kinect sensors with orthogonal viewing angles. They first identified the occluded body joints in the initial depth image and then replaced them with the corresponding joints obtained from the second view while considering certain connectivity constraints. Irrespective of the goal to complete occluded body parts using bi-view depth videos, this approach can also improve the accuracy of detecting visible joints in both videos. Relative calibration and temporal synchronization between depth sensors are prerequisites for the success of this method. Cai et al. [28] also proposed a different method for extracting skeletal joint points. After applying a median filter to reduce the noise in the depth images, they extracted edges to separate the human border from its background. In this method, morphological filters were used in depth images to reveal the initial skeletal structure of a human. Then, the positions of the joints were extracted by using knowledge about the relative size of the body’s skeletal components and fitting the triangle with the maximum area to the curves achieved in the previous step. This method also faces challenges, like the method by Shoton et al. [26], when dealing with occlusions of the skeletal structure. Kuang et al. [29] utilized a two-level threshold approach to separate the overlapping parts of the human body in the depth image. Subsequently, they merged the skeletal points extracted from each threshold stage to form a complete skeletal structure. Li et al. [30] developed a post-processing method to refine the skeletal structure that was extracted using the method by Shoton et al. [26]. They utilized knowledge about the relative distances between the body’s joints to prevent noise in the skeletal structure. In this paper, we utilized the method proposed by Shoton et al. [26] to determine the three-dimensional position of the body’s joints, available in Microsoft Kinect SDK (https://www.microsoft.com/en-us/download/details.aspx?id=44561, last accessed on 26 September 2024).

On the other hand, some studies have concentrated on recognizing motion or actions by analyzing 3D skeletal sequences [1,6,7,8,10,12,13,24,31,32,33,34,35,36,37]. Some works focused on using contact or wearable devices to measure the raw data of human actions [38]. On the other hand, remote measurements of 3D human skeletal sequences are an alternative data source, and some developed methods using such data sources are reviewed in more detail hereafter.

Wang et al. [31] proposed a method using data mining and machine learning to identify actions in 3D skeletal sequences obtained from depth images. In this approach, Local Occupancy Pattern (LOP) and temporal features extracted from the Fourier temporal pyramid are utilized to differentiate and categorize actions. Zhao et al. [32] refined the sequence of 3D skeletal points based on viewing angle and anthropometry. They then measured similarities with the training data using the Dynamic Time Warping (DTW) method. In this method, it is possible to tag different gestures in real time. However, it may face challenges when processing long sequences. Wang and Wu [33] also used the Learning Maximum Margin Temporal Warping method to detect motion in depth videos. This approach was developed to address the temporal asynchrony of actions and inconsistencies in sequence duration when comparing depth videos. Kerola et al. [34] identified 3D key points in both individual skeleton frames and integrated 3D skeletal points over time. They then utilized the Spectral Graph Wavelet Transform (SGWT) method to create descriptors for classifying human activities in depth videos. In this study, the classifiers used were a combination of PCA and SVM. Chiang et al. [35] utilized Kinect sensors to monitor and provide feedback to patients performing intervention or rehabilitation exercises. They simultaneously captured the 3D skeletal nodes using both a Kinect sensor and a motion capture (MOCAP) system during the training stage. They then trained a Gaussian process regression model to increase the reliability of their measurements. In this method, live-captured Kinect sequences were divided into subsequences, and each subsequence was compared with its corresponding reference subsequence. Pang and Liang [1] employed relative features among skeletal joint points to extract temporal features. They utilized a modified version of the DTW algorithm to compare feature vectors to measure their similarity. Salim et al. [36] also used Support Vector Machine (SVM) and Random Forest techniques to classify 14 human activities by extracting 46 features from human body joints captured with the Kinect sensor. Pham et al. [24] also designed two neural networks entitled Attention-enhanced Adaptive Graph Convolutional Network with Feature Fusion (FF-AAGCN) and Double-feature Double-motion Network (DD-Net) for labeling nine different physical exercise motions. The results and data used in this research have also been used in the evaluation of the proposed method in this article. Li et al. [36] proposed a method for the recalibration of Memory Attention Networks (MANs) and deployed the Temporal Attention Recalibration Module (TARM) and Spatio-Temporal Convolution Module (STCM) for skeleton-based action recognition. They showed that MANs can improve the performance of action recognition. However, their need for huge data in the training process is a common limitation of the deep learning approaches. Irrespective of the wearable devices used in [38] for action recognition, Haghi et al. used three neural networks for human activity recognition in approximately real time. Whereas the high speed in response to the trained network can be achieved, the necessity of numerous training data, overfitting problems, and the time of training are the main limitations of artificial intelligence (AI) methods. Sideridou et al. [25] also developed a smartphone application based on kinematic machine learning to recognize the performance quality of exercise movements and guide people to perform them correctly by writing some messages on the application screen. Extracting temporal and frequency domain features from the time series of some angles in the human skeletal structure has been their data source for separating and evaluating exercises. They also did not try to assess the performance quality of the exercises with the scores given by a coach.

Generally, the main sources of data in human motion recognition are the direct usage of the time series of the skeletal nodes’ motions or extracting features from them [25,32]. The phase differences or differences in execution speeds among the recorded exercise movements are the challenges when comparing the time series. Synchronization and phase removal are the necessary preprocessing steps before comparing time series, which has been done in different ways so far [25,32]. The use of mutual information descriptors has not been utilized to mitigate these issues when comparing the time series of skeletal nodes. On the other hand, the development of an approach for scoring the physical exercises’ performance compatible with the coach’s decision was not addressed in previous research. In this paper, we proposed a method to discriminate physical exercises and score them in terms of correctness in performing through the sequences of the human skeletal structures acquired by the Kinect depth sensor. In this method, the significant descriptors that represent distinct relationships between temporal features extracted from the skeletal structure for each physical exercise are used as bases in labeling and scoring. The advantage of this method is the independence of labeling to the temporal contraction or expansion in performing physical exercises. In the proposed method, we first extract several temporal features from the 3D skeletal joint points and then generate descriptive vectors based on the mutual information (MI) among the extracted temporal features. Each physical exercise is examined by its responses to different adaptive Constrained Energy Minimization (CEM) kernels that are specifically generated for all classes to find its label. The adaptive CEM kernels are generated by distinct components of MI descriptors for each class. The score for each labeled exercise is calculated using a simple geometrical metric that compares cross-correlation descriptors of labeled exercises among the selected training samples.

## 2. Materials and Methods

Action recognition and scoring are necessary in various applications such as interactive classrooms, medical rehabilitation, and controlling computer games [1,2,3,6,7]. Another application of action recognition is interactive dance or physical exercises taught through game consoles and computers, which is the subject of this research. The proposed method in this paper tries to first recognize the type and then the score of physical exercises performed in front of the Kinect sensor by comparing them to the correct samples. The flowchart in Figure 2 shows the detailed procedure of the proposed method. In this flowchart, dashed arrows indicate the process of using the training data, while solid line arrows illustrate the process of using the proposed method to recognize and score unlabeled exercises.

The proposed method consists of eight steps. The first step involves extracting 3D skeleton points in each frame of the depth videos. Since we used the method described by Shotton et al. [26] for extracting 3D skeleton joint points in depth frames, we opt not to elaborate on this method and refer the readers to the cited source for further information. In the second step of the proposed method, 36 instantaneous features are extracted from skeletal points in each frame, including 26 angular features and 10 features based on Normalized Difference Distance Indices (NDDIs). In the third step, the pairwise relative relationship among the extracted features in specific time intervals is quantified through MI and Pearson’s Correlation Coefficient (CC) indices to form two 630-component descriptor vectors. The MI descriptor vectors are utilized in the labeling process, while the CC descriptors are employed in the scoring process. The CC is utilized in many applications to study the linear dependency between two variables, e.g., see [39,40] for its definition and interpretation of its value ranges. Therefore, in the fourth step, the most distinctive components of the MI descriptor vectors are selected for discriminating each exercise from the others. In the fifth step, the adaptive target detector kernels are independently estimated for all classes through the CEM method. Then, in the sixth step, the kernels’ response is calculated for an unlabeled sequence of 3D skeleton motion to find the appropriate label. The labels of unknown sequences are determined based on both the winning response of the adaptive CEM kernels and the analysis of their CC descriptors. The entire 3D skeletal video is smoothly classified by repeating the fifth and sixth steps for all the sliding windows of a sequence. In the final step of the proposed method, the performance of the winning exercise is evaluated by comparing it with the ideal and incorrect performances using geometric measures calculated from the CC descriptors. Each step of this method is described in detail in the following subsections.

### 2.1. Angular and NDDI Features Extraction

Considering $SN\left(t\right)={\left[p_{1}p_{2},\ldots ,\;p_{20}\right]}^{T}$ as the 3D positions vector of skeletal nodes extracted from the depth frame at moment t, where $p_{i}$ is the 3D position of the i th node (Figure 1), the angular feature and Normalized Difference Distance Index at this moment are calculated through Equations (1) and (2), respectively.
(1)$$\alpha_{t}\left(p_{i},p_{v},\;p_{j}\right)=\arccos\left(\frac{d^{2}\left(p_{i},p_{v}\right)+d^{2}\left(p_{j},p_{v}\right)-d^{2}\left(p_{i},p_{j}\right)}{2d\left(p_{i},p_{v}\right)d\left(p_{j},p_{v}\right)}\right)$$
(2)$${NDDI}_{t}\left(p_{i},p_{j},p_{m},\;p_{n}\right)=\frac{d\left(p_{i},p_{j}\right)-d\left(p_{m},p_{n}\right)}{d\left(p_{i},p_{j}\right)+d\left(p_{m},p_{n}\right)}$$
where $\alpha_{t}\left(p_{i},p_{v},\;p_{j}\right)$ is the famous cosine formula for a triangle, and $d\left(p_{i},p_{j}\right)$ represents the Euclidean distance between two 3D nodes, i and j. In Equation (1), parameter *v* represents the vertex point of the angle. The angular features range from 0 to 180 degrees, and the NDDIs’ range is [−1,1].

In this research, 36 features are extracted at each moment from the sequence of skeletal nodes, as shown in Table 1. Our goal was to design these features so that they could effectively be used to distinguish among the exercises. The numbering of the skeletal nodes in Table 1 corresponds to Figure 1. After generating the temporal features $f_{1}\left(t\right)$ to $f_{36}\left(t\right)$ as per Table 1, the next step involves quantifying all their pairwise relationships, detailed in the following subsection.

All the features presented in Table 1 can simply be visualized in Figure 1 by drawing the connections based on the node numbers. In general, the combinations of constructing angular features and NDDIs were selected based on the visual analysis of the trajectory of the skeletal nodes during exercise movement executions.

### 2.2. Generating the CC and MI Descriptors

Considering the time interval $\left[t_{1},\;t_{2}\right]$ and the time interval between successive frames (dt) being equal to the frames per second (fps) of the depth video, a vector, as shown in Equation (3) for i=1,…, 36, can be generated for each of the features listed in Table 1.
(3)$$F_{i}\left(t_{1},t_{2}\right)={\left[f_{i}\left(t_{1}\right),\;f_{i}\left(t_{1}+dt\right),\;f_{i}\left(t_{1}+2dt\right),\ldots ,f_{i}\left(t_{2}-2dt\right),f_{i}\left(t_{2}-dt\right),f_{i}\left(t_{2}\right)\right]}^{T}$$

As seen in Equation (3), the number of extracted components for each of the temporal features listed in Table 1 is a function of the length of the sequence ($\Delta t=t_{2}-t_{1}$) and the fps of the depth videos. Therefore, alterations in the duration of the 3D skeleton sequence can influence both the length and the signature recorded within each feature vector. In the third step of the proposed method, we tried to produce descriptors with less dependence on the feature vectors’ length by quantifying their pairwise relative relationship. The advantage of this approach lies in its ability to mitigate the effects of temporal shifts and drift during the matching process between sequences of 3D skeleton motions. We employed the well-known cross-correlation (CC) coefficients, as outlined in Equation (4), and the mutual information (MI) index, detailed in Equation (5), to compute relative pairwise descriptors among the feature vectors, as described in Equation (3).
(4)$$CC\left(F_{i},F_{j}\right)=\frac{\sigma_{F_{i},\;\;F_{j}}}{\sigma_{F_{i}}\sigma_{F_{j}}}\;\;\;i,j=1,2,\ldots ,\;36\;\mathrm{and}\;i\neq j$$
(5)$$MI\left(F_{i},F_{j}\right)=E\left(F_{i}\right)+E\left(F_{j}\right)-JE\left(F_{i},\;F_{j}\right)\;\;\;i,j=1,2,\ldots ,\;36\;\mathrm{and}\;i\neq j$$

In Equation (4), $\sigma_{F_{i},\;F_{j}}$ is the covariance between the feature vectors *F_i_* and *F_j_*, and $\sigma_{F_{i}}$ and $\sigma_{\;F_{j}}$ are their standard deviations [41]. In Equation (5), E and JE are the well-known entropy and joint entropy operators, respectively [42]. In probability theory and information theory, the mutual information (MI) of two random variables is a measure of the mutual dependence between the two variables. More specifically, it quantifies the amount of information obtained about one random variable by observing the other random variable.

Given the 36 features extracted in Table 1, a total of 630+630=1260 MI and CC descriptors could be generated for each sequence of 3D skeletal nodes, independent from of the length of its feature vectors. In the estimation of the MI descriptors, the known range of the features is quantized to increase the computational speed. To do so, the known range of angular indices is quantized into 10-degree intervals, and the range of NDDI is divided into twenty parts.

### 2.3. MI Descriptor Selection for Each Exercise Motion

In the proposed method, for each physical exercise, a separate set of discriminating MI descriptors is selected. The descriptor selection strategy is designed in a way that finds the descriptors that can discriminate each exercise motion from the others.

Given $MID_{i,j}={\left[MI_{i,j,1},MI_{i,j,2},\ldots ,MI_{i,j,630}\right]}^{T}$ as the descriptor vector of the j th training sample of the i th physical exercise, the discriminative potential of each of its components to separate the i th exercise can be quantified by Equation (6).
(6)$${DI}_{i,k}=\frac{\left|{\bar{MI}}_{i,*,k}-{\bar{MI}}_{\neq i,*,k}\right|}{STD\left({MI}_{i,*,k}\right)+STD({MI}_{\neq i,*,k})}\;\;k=1,2,\ldots ,\;630$$
where ${\bar{MI}}_{i,*,k}$ and $STD\left({MI}_{i,*,k}\right)$ are the average and standard deviation of the k th component of the MID vectors for all the training examples of the i th, respectively. Also, ${\bar{MI}}_{\neq i,*,k}$ and $STD({MI}_{\neq i,*,k})$ are the average and standard deviation, respectively, of the k th component of the MID vectors for all training samples except the i th exercise. The symbol $\left|\cdot \right|$ in this equation is the absolute operator. According to the Equation (6), when the ${\bar{MI}}_{i,*,k}$ of the i th exercise has had a large difference in ${\bar{MI}}_{\neq i,*,k}$ of the other exercise motions, the values of the DI index will be large. In this equation, the STD values act as emphasizers for the selection of descriptors with lower deviations.

The DI is calculated for all 360 components of the MID vector for each physical exercise, and then, they are sorted (prioritized) in descending order. In the following, if there are no significant changes in prioritization, an attempt is made to refine the order of the indicators by considering the periodicity in the sign of the DI numerator. This is done to reduce the selection of descriptors with similar discrimination behavior. After possible refinement in the prioritization of descriptors, the components corresponding to a certain number (nD) of DI s with higher priority among the MID vector components are selected as the set of distinguishing descriptors for each exercise motion.

### 2.4. Estimation of the Adaptive CEM Kernels

Constrained Energy Minimization is a well-known target detection method in which identifying a detector kernel vector ($\vec{H_{q}}$) for each target quantifies the possible presence of the target in an unknown descriptors vector $\vec{D_{q}}\left(t_{1},t_{2}\right)$ by its response that is estimated through a simple dot product, as represented in Equation (7). Here, the targets will be exercising motions. In the CEM method, $\vec{H_{q}}$ is estimated in such a way that the sum of the squares of the detector responses is minimized for other descriptor vectors, except the target [42].
(7)$$\;\;{TR}_{q}\left(t_{i},t_{j}\right)=\vec{H_{q}}\left(t_{i},t_{j}\right)\cdot \vec{D_{q}}\left(t_{i},t_{j}\right)\;,\;\;\;t_{j}>t_{i}$$

In Equation (7), $\vec{H_{q}}\left(t_{i},t_{j}\right)$ represents the detector kernel vector associated with the target (q) over the time interval $\Delta t=[t_{i},t_{j}]$ within the video footage of 3D skeletal nodes. Similarly, $\vec{D_{q}}\left(t_{i},t_{j}\right)$ and ${TR}_{q}\left(t_{i},t_{j}\right)$ denote the vector of selected descriptors and the detector response corresponding to the target (q) during the same interval, respectively. Additionally, the notation “⋅” signifies the dot product operation.

According to Equation (6), in the proposed method, for each unlabeled skeletal 3D nodes sequence between $\left[t_{i},t_{j}\right]$ and each physical exercise (target), a detector kernel is adaptively identified according to the corresponding target training samples and the elapsed sequences $\left[t_{0},t_{j}\right]$. To do so, selected descriptors for each target of all the training samples and the elapsed sequences will be needed. Given the set $K_{q}=\left\{k_{q,1},k_{q,2},\ldots ,k_{q,nD}\right\}$ containing the index numbers of the selected MI descriptors related to exercise q, ${\vec{TSD}}_{i,j,k\in K_{q}}={\left[{MI}_{i,j,k_{q,1}},{MI}_{i,j,k_{q,2}},\ldots ,\;{MI}_{i,j,k_{q,nD}}\right]}^{T}$ is a vector containing the selected MI s of the q th exercise for the j th training sample of the i th exercise. On the other hand, $\vec{D_{q}}\left(t_{i},t_{j}\right)={\left[{MI}_{k_{q,1}}\left(t_{i},t_{j}\right),{MI}_{k_{q,2}}\left(t_{i},t_{j}\right),\ldots ,\;{MI}_{k_{q,nD}}\left(t_{i},t_{j}\right)\right]}^{T}$ is also the vector of the selected MI s for the q th physical exercises in the time interval $\left[t_{i},t_{j}\right]$ from the sequence of unlabeled 3D skeletal nodes. Accordingly, the adaptive detector kernel of the physical exercise q will be estimated through Equation (8) in the CEM method.
(8)$$\vec{H_{q}}\left(t_{i},t_{j}\right)=\frac{R_{q}^{-1}\left(t_{i},t_{j}\right)\times g_{q}}{g_{q}^{T}{\times R}_{q}^{-1}\left(t_{i},t_{j}\right){\times g}_{q}}$$

In Equation (8), $R_{q}(t_{i},t_{j})$ is the adaptive covariance matrix of the training samples and the elapsed sequences for the q th target selected descriptors (Equation (9)). This matrix was designed to simultaneously be affected by both the training samples and the elapsed depth videos as the background context. Vector $g_{q}$ is the representative vector of target q containing the mean values of its training samples of the selected descriptors (Equation (10)).
(9)$$\begin{array}{c} R_{q}\left(t_{i},\;t_{j}\right)=\frac{1}{nT}\sum_{i=1}^{nT}\frac{1}{n_{i}}\sum_{j=1}^{n_{i}}{\vec{TSD}}_{i,j,k\in K_{q}}\times {{\vec{TSD}}^{\;T}}_{i,j,k\in K_{q}}\\ +\frac{dt}{t_{i}}\int_{t=0}^{t_{i}}\vec{D_{q}}\left(t,\;t+∆t\right)\times {\vec{D_{q}}\left(t,\;t+∆t\right)}^{T}dt \end{array}$$
(10)$$g_{q}=\frac{1}{n_{q}}\sum_{j=1}^{n_{q}}{\vec{TSD}}_{q,j,k\in K_{q}}$$

In Equations (9) and (10), nT is the total number of physical exercises (targets), and $n_{q}$ is the number of training samples of the q th exercise. Variable ∆t is the time interval for separating the sequence of 3D skeleton nodes in the process of labeling the video obtained from physical exercises. Also, dt is the sliding step of the time interval for the successive generation of vectors $\vec{D_{q}}\left(t_{i},t_{j}\right)$ to be used in the process of labeling and scoring. The minimum value for dt can be set equal to the fps of the depth videos taken by the Kinect sensor.

From a theoretical point of view, the covariance matrix in the CEM method, $R_{q}\left(t_{i},t_{j}\right)$, should represent the statistical distribution of the background data [43,44]. Two different types of background motions might be present in the recorded depth videos when attempting to recognize a target exercise: irrelevant targets or non-target motions. Hence, the covariance matrix of the CEM kernels was designed to have a balanced effect from both types of backgrounds. Accordingly, the first nominal of Equation (9) is the target affected part of the covariance matrix, which is simply reconstructed by selected descriptors of the training samples. The second nominal of Equation (9) is the adaptive part of $R_{q}\left(t_{i},t_{j}\right),$ which reconstructs the statistical characteristics of the elapsed exercise motions (non-target backgrounds), before investigating $\vec{D_{q}}\left(t_{i},t_{j}\right)$. This will cause a better representation of the samples’ distribution in the descriptor space and reduce the possibility of the singularity in the $R_{q}^{-1}\left(t_{i},t_{j}\right)$ estimation when finding detector kernels.

### 2.5. Labeling of Unknown Sequences

After selecting the discriminating descriptors and identifying the CEM detector kernels for every physical exercise, as outlined in the flowchart presented in Figure 2, the procedure for labeling unknown sequences will commence, following the solid arrows depicted. This procedure can be easily executed on a long sequence of depth videos containing extracted 3D skeletal nodes. To achieve this, the entire sequence of 3D skeleton nodes must be divided into recognized ∆t intervals using a sliding temporal window. Following this, by extracting the features listed in Table 1 for each frame, the selected descriptors for each targeted exercise will be produced in the form of $\vec{D_{q}}\left(t_{i},t_{j}\right)$ (q=1,2,…,nT) vectors. Then, the response vector of the CEM detector kernels, Equation (6), is calculated as represented in Equation (11).
(11)$$TR\left(t_{i},t_{j}\right)={\left[{TR}_{1}\left(t_{i},t_{j}\right),\;{TR}_{2}\left(t_{i},t_{j}\right),\;\ldots ,\;{TR}_{nT}\left(t_{i},t_{j}\right)\right]}^{T}$$

In the proposed method, the labeling of each sequence is performed through the simultaneous provision of two criteria. In this process, the initial label of each sequence $L\left(t_{i},t_{j}\right)$ is identified through the maximum value of the $TR\left(t_{i},t_{j}\right)$ vector; see Equation (12).
(12)$$L\left(t_{i},t_{j}\right)=argmax\left(TR\left(t_{i},t_{j}\right)\right)L\in \left\{1,2,\ldots ,nT\right\}$$

Then, the initial labeling should be confirmed by comparing the CC descriptors as the second criterion. To accomplish this, the Pearson cross-correlation between the CC descriptor vectors of the initially labeled sequences and the training data representing the winning label must exceed 50%.

### 2.6. Scoring on the Performance of Exercises

By finding the label of each sequence, its performance quality is scored by comparing the CC descriptors to their corresponding training samples. However, due to the possible inconsistency between semantic scoring assigned by the exercise referees and similarity metrics between CC descriptors, finding a conversion between the two will be necessary. The process of scoring requires both labeled and scored training samples for each class. Each labeled sequence is scored by its descriptor vector’s location among the corresponding descriptors of the scored training samples. Since the imperfect executions of an exercise motion can alter its descriptor vectors’ values, some of the training samples should be selected from the imperfect executions. The concept behind the scoring process in our method is simply visualized in Figure 3. We have shown the descriptors space (630D) in a 3D space for simplicity in comprehension.

The stars in Figure 3 schematically represent the position of the vectors containing CC descriptors of the training samples related to the winning label of an exercise motion sequence. The red star is related to the position of the training sample with a perfect performance (score of 100), and the blue stars show the training samples with imperfect executions (scores less than 100). Here, the score of each star is displayed with $S_{b}=100$ and $S_{w_{1}}$,$S_{w_{2}}$, and $S_{w_{3}}$. As seen, three imperfect training samples have been drawn in Figure 3; however, the number of which can be different according to the available training samples in the scoring process. The yellow cross is the position of an exercise descriptors vector with the same label of drawn stars which score is unknown. The perpendicular distance of the yellow cross to the lines connecting each imperfect training sample to the perfect one is displayed with values $v_{1}$ to $v_{3}$. During the scoring process, the line with the smallest perpendicular distance to the yellow cross is chosen as the reference for scoring. According to Figure 3, two distances: “d” (the distance between the vertical projection of the yellow cross and the red star position) and “D” (the distance between the red star and the associated blue one from the selected line) will be used to calculate the score. The mentioned d (vertical projection of (a) the vector connecting the red star to the yellow cross on (b) the unit vector that is colinear with the vector connecting the winner line endpoints) is calculated by simply using the dot product. Therefore, it can be either a positive or negative sign. Equation (13) shows how to calculate the score for the example of Figure 3. This equation is designed based on a simple linear interpolation.
(13)$$Score=\left\{\begin{array}{ll} \;\;\;\;0 & \;\;\;d>\frac{D}{100-S_{w_{3}}}\\ 100 & d<0\;\;\;\;\;\;\;\;\;\;\;\;\;\;\;\;\;\;\;\;\\ 100-\left(d/D\right)\left(100-S_{w_{3}}\right) & Otherwise \end{array}\right.\;$$

In Equation (13), $S_{w_{3}}$ is the score of the selected training sample with a score of less than 100 in Figure 3, and the line connected to it has the smallest distance to the yellow cross. Therefore, in other similar situations, $S_{w_{3}}$ will be replaced by the selected training sample score. According to this figure, if the sign of “d” is negative, it shows the yellow cross lies before the red star (perfect execution of exercise), and its score is supposed to be 100.

## 3. Results and Discussion

The proposed method has been evaluated in two datasets. The first dataset was created by the authors, and the second one was the reference data used in [24]. The dataset produced by us included nine exercise motions, and each movement was performed 10 times, for 90 repetitions in total, in front of the Kinect sensor. Nine short videos are available in the Appendix A that show the performance of each exercise movement.

In creating the first dataset, three experienced physical coaches (two men and a woman) were engaged to execute each exercise correctly, repeating each action twice. In total, eight participants took part in the data collection, and the details are presented in Table 2.

The total length of the depth video in the first dataset was 245 min, of which a total of 116 min was dedicated to target exercises, and at other times, unrelated motions were performed in front of the Kinect sensor. The Kinect sensor was mounted at a suitable distance and orientation when taking depth video from the players. These conditions included a straight-ahead and full-height vision of the player’s body in the captured depth videos. The general environmental conditions of recording depth images were also the same as the computer games, controlled by Kinect. Generally, the presence of intense light in the environment deteriorates the matching process in the Kinect depth sensor, and the environment and players must not be covered with light-absorbing (dark) materials.

The second dataset also consisted of 297 labeled distinct sequences of 3D skeletal nodes that were categorized into nine different exercise performances. The labeling performance of the proposed method has been evaluated at two levels. At first, by changing its adjusting parameters, their effects were investigated on the results of the labeling process. At the second level, the performance of the proposed method was compared with the method developed in [24], which dataset has been used in our evaluations. The adjusting parameters of the proposed method are introduced in Table 3.

According to Table 3, a total of 100 modes of implementation of the proposed method with different adjusting parameters were designed to evaluate their effects on the labeling process. In each set of adjusting parameters, the accuracy of the labeling process was measured by indices of the overall accuracy and the Kappa coefficient extracted from the confusion matrix. To do so, the true labels of the physical exercises were assigned throughout the whole dataset and then compared with the predicted labels by the proposed method. The graphs presented in Figure 4 show the overall accuracy and Kappa coefficients of all 100 different sets of adjusting parameters in implementing the proposed method.

According to Figure 4, up to 30% differences in overall accuracies were observed when changing the adjusting parameters, which shows the importance of finding the best setting to achieve optimal results. Another inferable result in these graphs is the small sensitivity of the results to the change in the sampling rate of the depth videos (fps), as if good labeling results can be achieved even with 4 fps videos. However, increasing the length of the video sequence used in the labeling process has led to a decreasing trend in overall accuracies. It seems the limited temporal length of each exercise performance and the possibility of their non-successive repeating in the recorded depth videos are the reasons for the decreasing trend of accuracies by increasing the length of the sequences. However, the sequences with lengths of 5 to 10 s did not have a significant change in decreasing the accuracies. Therefore, choosing such lengths for the sequences is recommended in the labeling process. Based on the obtained results, the other inferred advantage of the proposed method could be the low number of descriptors required for recognition of each physical exercise. As shown in Figure 4, the best results have been obtained with a minimal selection of descriptors (e.g., two descriptors for each exercise). This can be one of the strengths of this method compared to the ones that use neural networks or many complex descriptors in the labeling process like [24,34]. Reducing the number of generated descriptors in the labeling process can directly impact the computational costs and facilitate the possibility of obtaining real-time solutions in low-cost processing units. From the functional transparency point of view, developing such a method with clear steps is more interpretable than the neural network approaches with the black box nature, irrespective of their pragmatic efficiencies. The confusion matrix of the best-achieved result of the proposed method in the first dataset is presented in Figure 5.

As can be seen in the confusion matrix reported in Figure 5, the main uncertainties were related to the unclassified class that has a large diversity. Some of the unclassified sequences were also related to the time interval that the sequence of skeletal nodes was not completely located at during the exercise movement execution.

Next, we compared the results obtained from the proposed method using the best setting parameters in the second dataset with the results of [24] implemented on the same dataset. In [24], two deep learned neural networks named FF-AAGCN and DD-Net were used to label the nine actions in the sequence of 3D skeletal nodes. The confusion matrices of the proposed method and the mentioned comparison methods are presented in Figure 6.

By comparing the confusion matrices in Figure 6, the competitive accuracies of the proposed method can be seen. Note that the number of training and test samples used to achieve these results in the proposed method was selected similar to the ones used in the comparison methods. These were 55% train and 45% test. This demonstrated that the proposed method, with its simple and comprehended steps, can do what a complex deep-learned neural network can.

Our next evaluation was dedicated to the scoring process. As described in Section 2.6, one CC descriptor vector related to the perfect execution of each class and at least one sample with imperfect execution (score less than 100) are needed in the scoring process. Therefore, the only adjusting parameter of the scoring process is the number of imperfectly executed training samples with scores of less than 100. In evaluating the scoring process, the role of this adjusting parameter has been measured.

After determining the label of each sequence of the 3D skeletal nodes, its score can be calculated through the training samples of the winner label. We could not find any reference dataset with coach scores for 3D skeletal sequences for cross-validation. Therefore, we exclusively utilized our carefully prepared dataset, in which each exercise performance had been scored by an expert coach. The scoring results of the proposed method with the coach grades could be easily compared. Note that, in this evaluation, only the sequences were used that were completely in the time frame of doing the exercise motions. In other words, if a sequence contained the execution of more than one exercise, it was considered a mixed sequence that was not used in the evaluation process of scoring performance. This was done to eliminate these mixed sequences which CC descriptors were not purely dedicated to a unique exercise motion. Obviously, scoring the mixed sequences will be ambiguous, and we do not have any truth values for their comparisons. The scatterplots represented in Figure 7 show the predicted scores compared to their correct values for the pure sequences of our dataset. Each scatterplot represents the results of the scoring process with a different number of imperfect training samples.

As seen in the scatterplots of Figure 7, increasing the number of imperfect training samples with scores less than 100 can be effective in improving the accuracy of the scoring process. However, there is no significant difference between the selection of three and five imperfect training samples. Therefore, it seems that the availability of at least three training samples with scores less than 100 can achieve favorable results in the scoring process. The slopes of the fitted line (near 1) and the R^2^ to each scatterplot indicate the low level of remaining bias in the scoring step. The Root Mean Square Errors (RMSEs) of the scatterplots represented in Figure 7 are also reported in Table 4.

In general, the proposed scoring method is a mapping of measurable quantities from the descriptive vectors of the sports coach’s linguistic scoring. The separation of the results reported in the graphs of Figure 7 into factors, such as the gender, height of the players, and the type of exercise movement, did not lead to meaningful and comparable results. This shows that, if the correct label is assigned to an exercise movement, the scoring process can be matched with the points obtained by a sports coach independently from these factors. The authors believe that the extracted features (angular and NDDI features) may have the potential to overcome these types of uncertainty due to their normalizing nature.

## 4. Conclusions

We proposed a method for continuously classifying and scoring the captured videos of physical exercises through the Kinect depth sensor. Our method was based on extracting relative descriptors between features extracted from 3D skeletal nodes. The labeling of video sequences was performed using adaptive CEM kernels and scoring through interpolation among CC descriptors. The results demonstrated that our method could achieve the proper accuracies in the labeling process, and its scoring results were close to ones derived by sports coaches. This idea can be used to develop an interactive computer game for game consoles with the aim of teaching dance or physical exercises. Essential changes that should be considered in this method when there is more than one depth sensor in capturing 3D skeletal nodes could also be considered in future investigations.

## Figures and Tables

**Figure 1 sensors-24-06713-f001:**
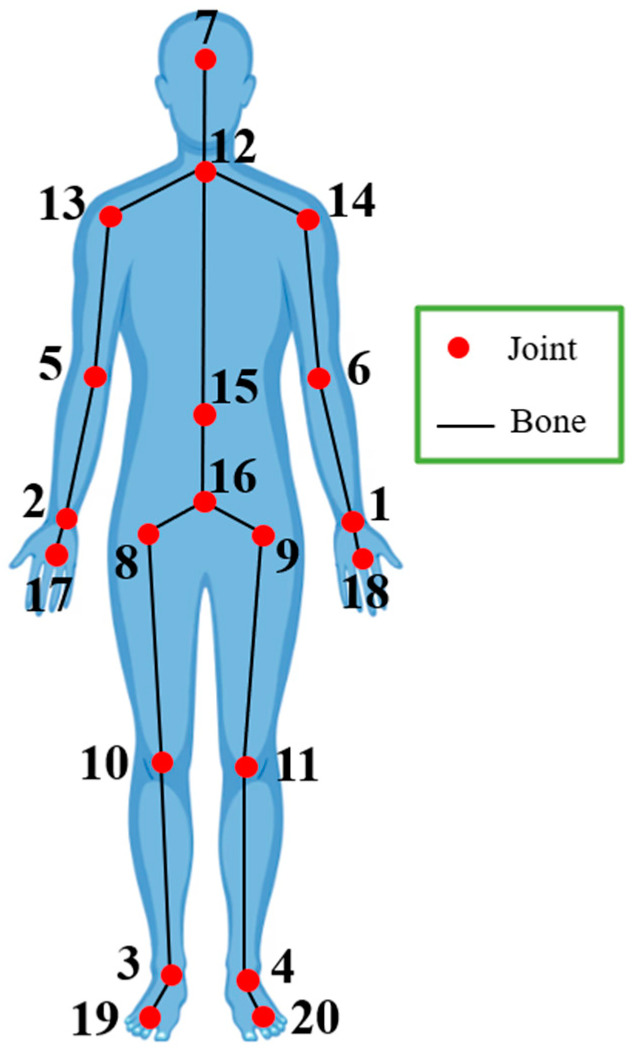
Skeletal joint points extracted from a human depth image.

**Figure 2 sensors-24-06713-f002:**
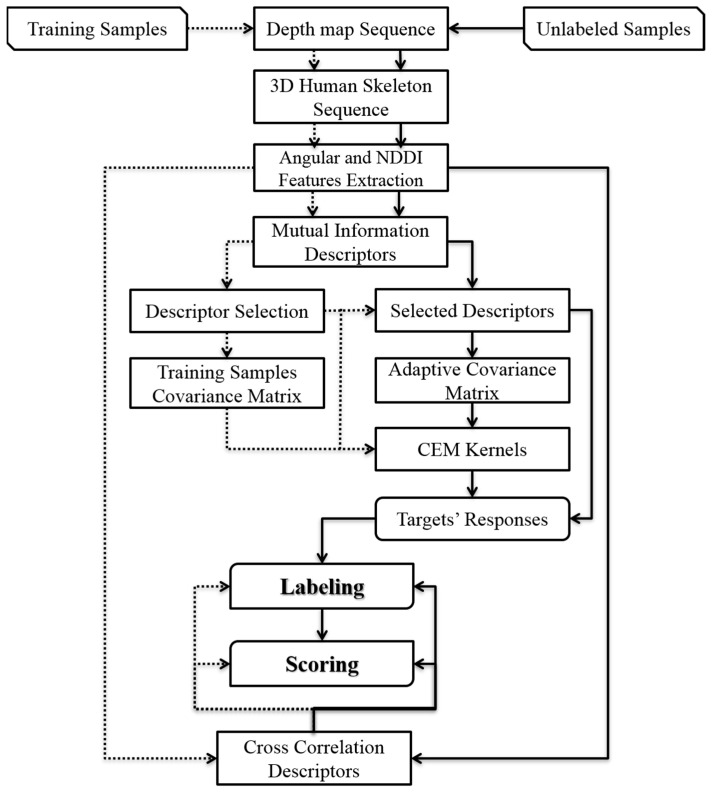
Flowchart of the proposed method for recognizing and scoring physical exercises.

**Figure 3 sensors-24-06713-f003:**
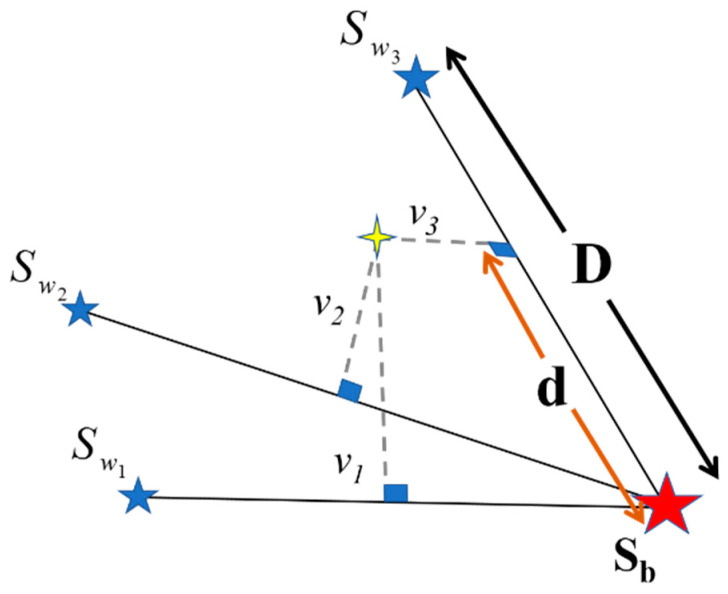
A schematic drawing of the scoring process in the CC descriptors space.

**Figure 4 sensors-24-06713-f004:**
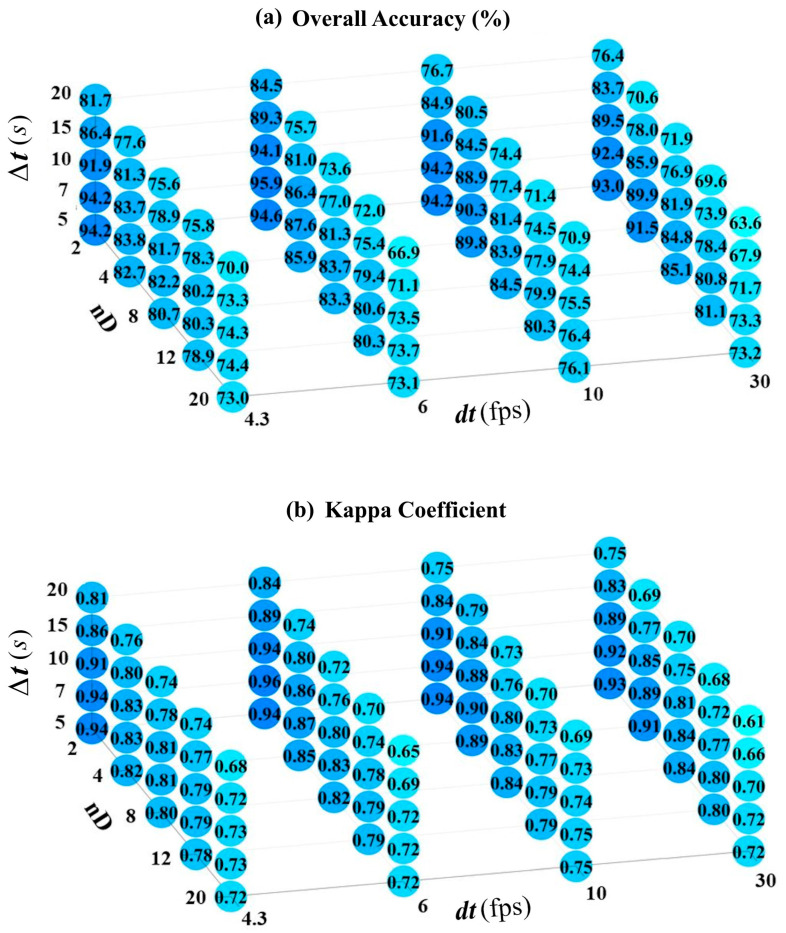
Overall accuracy and Kappa coefficient of the labeling by changing the adjusting parameters.

**Figure 5 sensors-24-06713-f005:**
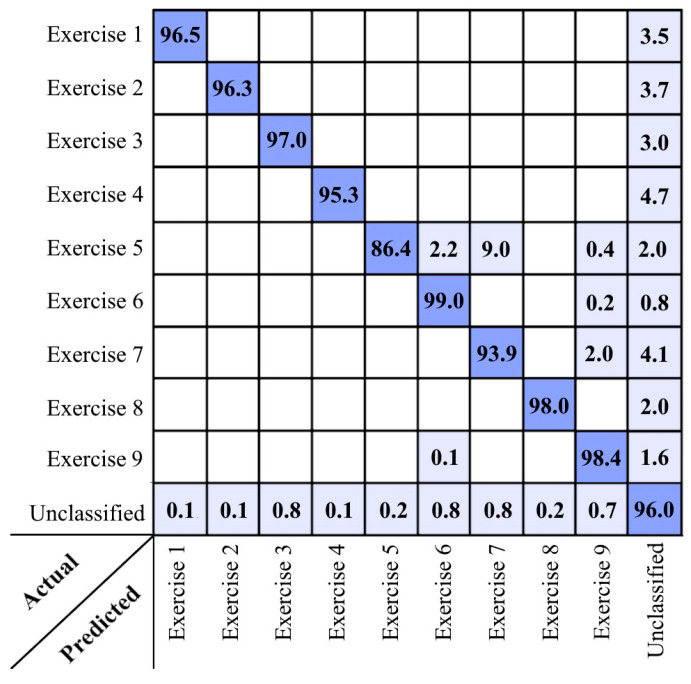
Confusion matrix of the best result achieved by the proposed method in the first dataset.

**Figure 6 sensors-24-06713-f006:**
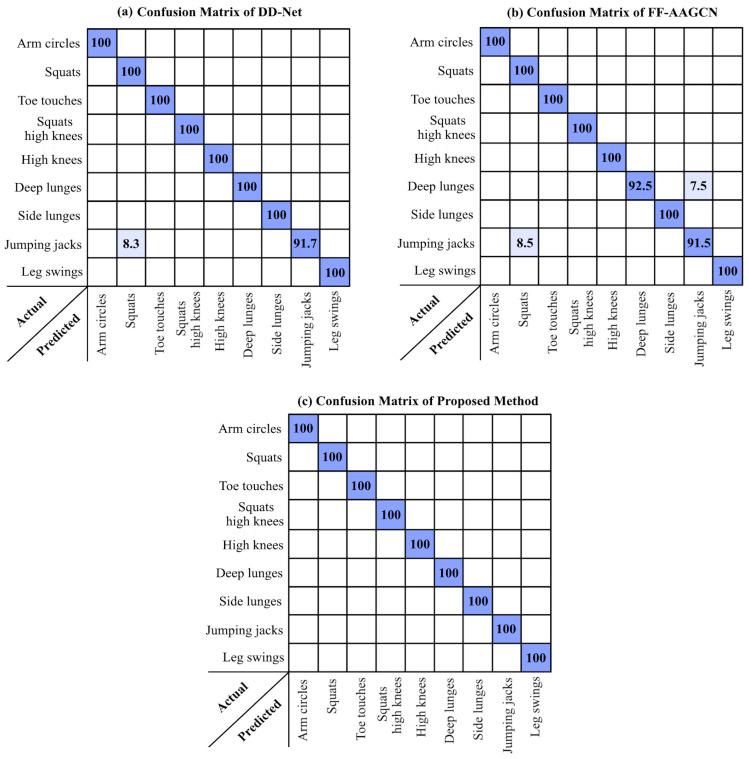
The confusion matrix of the proposed method and the results obtained from [24] in the second dataset.

**Figure 7 sensors-24-06713-f007:**
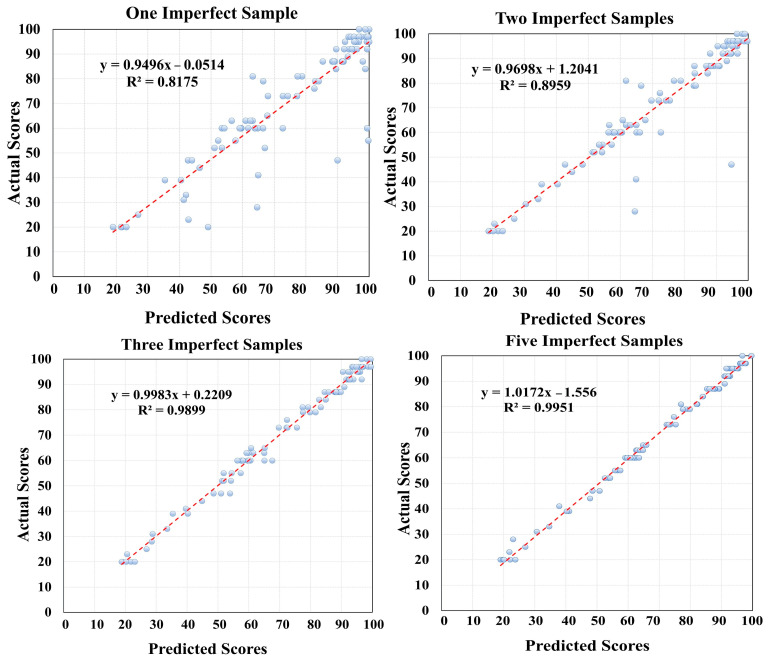
Scatterplots of the scoring process for different numbers of imperfect training samples.

**Table 1 sensors-24-06713-t001:** Angular features and NDDIs extracted from skeletal nodes.

$0^{\circ }<\alpha_{t}\left(p_{i},p_{v},p_{j}\right)<180^{\circ }$								$-1<NDDI_{t}\left(p_{i},p_{j},p_{m},p_{n}\right)<1$				
f	i	v	j	f	i	v	*j*	f	i	j	m	n
$f_{1}\left(t\right)$	2	5	13	$f_{14}\left(t\right)$	1	16	4	$f_{27}\left(t\right)$	1	4	2	3
$f_{2}\left(t\right)$	1	6	14	$f_{15}\left(t\right)$	2	16	4	$f_{28}\left(t\right)$	1	2	4	3
$f_{3}\left(t\right)$	3	10	8	$f_{16}\left(t\right)$	1	16	3	$f_{29}\left(t\right)$	6	11	5	10
$f_{4}\left(t\right)$	4	11	9	$f_{17}\left(t\right)$	5	12	16	$f_{30}\left(t\right)$	6	5	11	10
$f_{5}\left(t\right)$	6	16	11	$f_{18}\left(t\right)$	6	12	16	$f_{31}\left(t\right)$	5	10	7	16
$f_{6}\left(t\right)$	5	16	10	$f_{19}\left(t\right)$	10	16	12	$f_{32}\left(t\right)$	6	11	7	16
$f_{7}\left(t\right)$	5	16	11	$f_{20}\left(t\right)$	11	16	12	$f_{33}\left(t\right)$	2	10	7	16
$f_{8}\left(t\right)$	6	16	10	$f_{21}\left(t\right)$	2	12	16	$f_{34}\left(t\right)$	1	11	7	16
$f_{9}\left(t\right)$	11	16	10	$f_{22}\left(t\right)$	1	12	16	$f_{35}\left(t\right)$	2	3	7	16
$f_{10}\left(t\right)$	6	12	5	$f_{23}\left(t\right)$	3	16	12	$f_{36}\left(t\right)$	1	4	7	16
$f_{11}\left(t\right)$	2	12	1	$f_{24}\left(t\right)$	4	16	12					
$f_{12}\left(t\right)$	3	16	4	$f_{25}\left(t\right)$	12	5	1					
$f_{13}\left(t\right)$	2	16	3	$f_{26}\left(t\right)$	12	6	2					

**Table 2 sensors-24-06713-t002:** Specifications and roles of the participants involved in generating the first dataset.

Role	Gender	Age (Years)	Hight (cm)	Weight (kg)
Coach	Female	27	167	57
Participant	Female	35	163	65
Participant	Female	45	160	70
Participant	Male	19	180	75
Participant	Male	26	185	88
Coach	Male	30	178	68
Participant	Male	42	179	90
Participant	Male	60	173	80

**Table 3 sensors-24-06713-t003:** Adjusting parameters used for evaluation of the proposed method.

Parameters	Description	Set Values
∆t	The sequence length	5, 7, 10, 15 and 20 s
dt	Depth video rate	30 fps, 10 fps, 6 fps and 4.3 fps
nD	Number of selected MI descriptors for each exercise	2, 4, 8, 12 and 20

**Table 4 sensors-24-06713-t004:** The RMSEs of the scoring step.

Number of Imperfect Samples	RMSEs (Points)
1	10.9
2	7.8
3	2.4
5	1.7

## Data Availability

The datasets used in this research will be available upon reasonable request.

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
