# Peer review of "Recognition and Scoring Physical Exercises via Temporal and Relative Analysis of Skeleton Nodes Extracted from the Kinect Sensor"

_sensors, 2024, doi:10.3390/s24206713_

Round 1
Reviewer 1 Report
Comments and Suggestions for Authors
The manuscript presents a novel approach to recognizing and scoring physical exercises using features extracted from the skeletal joints captured by a Kinect sensor. The authors have developed a method that relies on pattern recognition for labeling and then scoring the exercises based on their performance accuracy, utilizing Cross Correlation (CC) coefficients in the scoring process. This is an interesting contribution to the field of human activity recognition (HAR) with potential applications in interactive systems, such as computer games, fitness monitoring, and rehabilitation.
Strengths:
1. The paper provides a clear description of the methodology, including the feature extraction, descriptor generation, and the use of Constrained Energy Minimization (CEM) kernels for exercise recognition.
2. The reported accuracy of 95.9% for the labeling process is quite impressive and indicates a robust system for identifying different types of physical exercises.
3. The inclusion of a scoring mechanism that employs CC coefficients to compare the performed exercises against reference samples is a valuable addition. It not only identifies the type of exercise but also evaluates the quality of the execution, which can be very useful for providing feedback in training or rehabilitation scenarios.
4. The use of both Mutual Information (MI) and Pearson's Correlation Coefficient (CC) to form descriptors adds depth to the analysis and enhances the discriminative power of the method.
Areas for Improvement:
1. A more detailed discussion of the limitations of the proposed method would strengthen the manuscript. Specifically, it should address how well the system performs across diverse users and under varying conditions, such as different lighting, backgrounds, and clothing.
2. To better contextualize this work's contributions, the authors should provide a comparative analysis with other state-of-the-art methods. This should include a comparison of recognition accuracy, computational efficiency, and the robustness of the scoring mechanism.
3. The dataset used for evaluation should be described in more detail, including the number of participants, the variety of exercises, and any variations in how these exercises were performed. Additionally, the authors could consider evaluating their method on publicly available datasets to allow for direct comparisons with other research. For instance, Figure 7 should provide a more comprehensive breakdown of the data, including the distribution of exercises, the demographics of the participants, and the specific metrics used to evaluate the performance.
4. Improving the quality of Figures 4 and 6 would make the manuscript more readable and engaging. Higher-resolution images and more evident labels would enhance the visual presentation of the results and help the reader better understand the key points being made.
Overall, the manuscript makes a solid contribution to the field and has the potential to advance the state of the art in HAR. With additional work to address the points above, the paper could be even more substantial and impactful.
Author Response
Dear reviewer,
Please find attached our response letter.
Thanks a lot!
Authors

Reviewer 2 Report
Comments and Suggestions for Authors
The authors proposed a framework to recognize human motion by considering a series of processes, such as gathering action from Kinect sensors, applying NDDI to extract features, computing CEM kernels, and scoring the exercise performance. The two datasets from the authors and paper [24] are used to demonstrate how the proposed framework works. However, the conduction of the literature review is not well described and several questions need to be addressed:
1. In the introduction, the review literature is not well comprehensive. Several recent papers like from the Journal of Sensors are not reviewed well. Several papers such as "Recognizing Human Activity of Daily Living Using a Flexible Wearable for 3D Spine Pose Tracking", "Towards Automating Personal Exercise Assessment and Guidance with Affordable Mobile Technology", etc. should be reviewed.
2. What are the current research gaps from previous literature? How can your method solve the research gaps? Why use Kinect sensors? Those detailed descriptions of strength motivation need to be addressed.
3. The first dataset collected by the authors is not clear. What are the exercises 1 to 9? Providing a few images with descriptions can be helpful.
4. The citations of equations are needed such that what is the equation name and source of equation 1 and other equations should be included in their sources and where the papers are from for the citation. For remaining eqs (5) to (10), the detailed physical meaning is required to explain what the value represents. In addition, titles 2.1 and 2.2 are replicated.
5. In Table 1, what are your criteria for pairing two joins in your 36 features? As they have a 20C2 combination. How do you pair two joins and decide they are the right features?
6. As the paper’s idea is from paper [25]. It still needs a paragraph to describe the methods of the paper [25] and how your paper adopts [25] your proposed framework.
7. In section 2.6, what is the yellow star? In eq. (13), when d is less than 0, the score is 100. Since the d is the distance between the vertical projection of the yellow cross and the red start position. Can the distance be a negative value? In addition, what is the condition of otherwise? In Figure 7, how do you predict the scores with any models?
8. In Table 2, the 60 fps is also a common frame for sensor detection. Why not include the 60 fps?
Author Response
Dear reviewer,
Attached please find our response letter.
Thanks a lot!
Authors

Round 2
Reviewer 2 Report
Comments and Suggestions for Authors
Please recheck the whole paper for grammar and typos for publication.